# Cycles of autoubiquitination and deubiquitination regulate the ERAD ubiquitin ligase Hrd1

Brian G Peterson[1], Morgan L Glaser[1], Tom A Rapoport[2], Ryan D Baldridge[1]*

[1]Department of Biological Chemistry, University of Michigan Medical School, Ann Arbor, United States; [2]Department of Cell Biology, Harvard Medical School, Howard Hughes Medical Institute, Boston, United States

**Abstract** Misfolded proteins in the lumen of the endoplasmic reticulum (ER) are retrotranslocated into the cytosol and polyubiquitinated before being degraded by the proteasome. The multi-spanning ubiquitin ligase Hrd1 forms the retrotranslocation channel and associates with three other membrane proteins (Hrd3, Usa1, Der1) of poorly defined function. The Hrd1 channel is gated by autoubiquitination, but how Hrd1 escapes degradation by the proteasome and returns to its inactive ground state is unknown. Here, we show that autoubiquitination of Hrd1 is counteracted by Ubp1, a deubiquitinating enzyme that requires its N-terminal transmembrane segment for activity towards Hrd1. The Hrd1 partner Hrd3 serves as a brake for autoubiquitination, while Usa1 attenuates Ubp1's deubiquitination activity through an inhibitory effect of its UBL domain. These results lead to a model in which the Hrd1 channel is regulated by cycles of autoubiquitination and deubiquitination, reactions that are modulated by the other components of the Hrd1 complex.

**\*For correspondence:**
ryanbald@med.umich.edu

**Competing interests:** The authors declare that no competing interests exist.

## Introduction

Proteins translocated into the endoplasmic reticulum (ER) undergo quality control, such that only folded proteins are moved through the secretory pathway. If a protein does not reach its native folded state, it is eventually retrotranslocated across the ER membrane, polyubiquitinated, extracted from the membrane, and degraded by the proteasome, a pathway called ER-associated protein degradation (ERAD) (for recent reviews, see *Berner et al., 2018*; *Preston and Brodsky, 2017*; *Wu and Rapoport, 2018*). Work in *S. cerevisiae* showed that substrates use four distinct ERAD pathways, depending on the localization of their misfolded domains. ERAD-L substrates contain misfolded domains in the ER lumen, ERAD-M substrates are misfolded within the membrane, ERAD-C substrates are membrane proteins with misfolded cytosolic domains, and ERAD-INM handles misfolded proteins in the inner nuclear membrane. These pathways use different ubiquitin ligases: ERAD-L and -M use the Hrd1 ligase, ERAD-C the Doa10 ligase, and ERAD-INM the Asi ligase complex (*Carvalho et al., 2006*; *Foresti et al., 2014*; *Huyer et al., 2004*; *Khmelinskii et al., 2014*; *Vashist and Ng, 2004*). These ligases are multi-spanning membrane proteins with cytosolic RING finger domains. Following polyubiquitination, all pathways converge at the Cdc48 ATPase (p97 or VCP in mammals) (*Bays et al., 2001*; *Jarosch et al., 2002*; *Rabinovich et al., 2002*; *Ye et al., 2001*). This ATPase cooperates with a cofactor (Ufd1/Npl4) to extract polyubiquitinated substrates from the membrane (*Stein et al., 2014*).

Among the ubiquitin ligases, the function of Hrd1 is best understood. Hrd1 forms a complex with three other membrane proteins (Hrd3, Usa1, Der1) (*Carvalho et al., 2006*; *Denic et al., 2006*; *Gardner et al., 2000*). Hrd3 is a single-spanning membrane protein with a large lumenal domain that interacts with substrates and Hrd1 (*Gauss et al., 2006a*; *Gauss et al., 2006b*). In the absence of

**eLife digest** Just like factories make mistakes when producing products, cells make mistakes when producing proteins. In cells, a compartment called the endoplasmic reticulum is where about one third of all proteins are produced, and where new proteins undergo quality control. Damaged or misfolded proteins are removed by a process called endoplasmic reticulum-associated degradation (ERAD for short), because if damaged proteins accumulate, cells become stressed.

One type of ERAD is driven by a protein called Hrd1. Together with other components, Hrd1 labels damaged proteins with a ubiquitin tag that acts as a flag for degradation. Hrd1 has a paradoxical feature, however. To be active, Hrd1 tags itself with ubiquitin but this also makes it more prone to becoming degraded. How does Hrd1 remain active while avoiding its own degradation?

To address this question, Peterson et al. forced budding yeast cells to produce high levels of 23 different enzymes that remove ubiquitin tags. One of these enzymes, called Ubp1, was able remove the ubiquitin tag from Hrd1, though it had not been seen in the ERAD pathway before. Further experiments also showed that Ubp1 was able to regulate Hrd1 activity, making Ubp1 a regulator of Hrd1 dependent protein quality control.

Without protein quality control, damaged proteins can contribute to various diseases. ERAD is a common quality control system for proteins, present in many different species, ranging from yeast to animals. Therefore, understanding how ERAD works in budding yeast may also increase understanding of how human cells deal with damaged proteins.

Hrd3, Hrd1 is strongly autoubiquitinated and rapidly degraded (*Gardner et al., 2000*). Usa1 is a double-spanning membrane protein that serves as a linker between Hrd1 and Der1 and facilitates the oligomerization of Hrd1 (*Carvalho et al., 2010*; *Horn et al., 2009*). It also has a ubiquitin-like (UBL) domain of poorly defined function; the UBL domain is dispensable for the degradation of ERAD substrates, but is required for the efficient degradation of Hrd1 in a *hrd3Δ* strain (*Carroll and Hampton, 2010*; *Vashistha et al., 2016*). Der1 is a multi-spanning protein required for ERAD-L, but not ERAD-M; it probably recognizes misfolded substrates in the ER lumen and facilitates their insertion into Hrd1 (*Knop et al., 1996*; *Mehnert et al., 2014*).

Recent results suggest that the Hrd1 ligase forms a protein-conducting channel (*Baldridge and Rapoport, 2016*). Overexpression of Hrd1 in *S. cerevisiae* cells bypasses the requirement for the other components of the complex, while all downstream components, such as the ubiquitination machinery and Cdc48 ATPase complex, are still needed (*Carvalho et al., 2010*). These results suggest that Hrd1 is the only essential membrane protein for a basic ERAD-L process. A cryogenic electron microscopy (cryo-EM) structure shows that the membrane-spanning segments of Hrd1 surround a deep aqueous cavity, supporting the idea that Hrd1 can form a channel (*Schoebel et al., 2017*). In vitro experiments further demonstrate that Hrd1 reconstituted into proteoliposomes allows a misfolded substrate domain to retrotranslocate across the lipid bilayer (*Baldridge and Rapoport, 2016*). This process requires autoubiquitination of Hrd1, leading to the suggestion that Hrd1 forms a ubiquitin-gated channel. The important autoubiquitination event occurs in the RING finger domain, as mutation of crucial lysines in this domain blocks retrotranslocation in vitro and ERAD-L in vivo (*Baldridge and Rapoport, 2016*).

If the Hrd1 channel is activated by autoubiquitination, how is Hrd1 spared from degradation and returned to its inactive ground state? Here, we identify Ubp1 as a membrane-bound deubiquitinating enzyme (DUB) that reverses the polyubiquitin modification of Hrd1 and allows Hrd1 to escape uncontrolled degradation. The Hrd1 partner Hrd3 serves as a brake for autoubiquitination, while the UBL domain of Usa1 attenuates Ubp1's activity, allowing Hrd1 autoubiquitination and activation. This delicate balance allows Hrd1 to undergo cycles of autoubiquitination and deubiquitination during ERAD.

## Results

### Ubp1 overexpression stabilizes Hrd1

Our previous experiments indicated that Hrd1 is autoubiquitinated in wild-type yeast cells (*Baldridge and Rapoport, 2016*). The protein is moderately stable, with a half-life of about 100 min. We therefore reasoned that overexpression of a DUB that reverses the modification of Hrd1 would increase the steady state levels of the ligase. We overexpressed 23 different DUBs in yeast cells that also express Flag-tagged Hrd1. The levels of Hrd1 were determined by immunoblotting with anti-Flag antibodies (*Figure 1A*). The strongest increase of Hrd1 levels was seen with Ubp1, even though several other DUBs were expressed at a higher level than Ubp1 (*Figure 1—figure supplement 1A*). Ubp1 overexpression had no effect on the levels of Hrd3 or Usa1 (*Figure 1—figure supplement 1B, C*).

A more sensitive analysis of Ubp1 function can be performed with *hrd3Δ* cells, in which Hrd1 autoubiquitination and degradation are greatly accelerated (half-life of about 30 min). Indeed, while Hrd1 levels are very low in *hrd3Δ* cells (*Figure 1B*; lane 3), Ubp1 overexpression increased Hrd1 to about the same level as in wild-type cells (*Figure 1B*; lane 8 versus 1). This increase was not seen when an enzymatically inactive mutant of Ubp1 (C110S) was overexpressed (*Figure 1B*, lane 13), indicating that the deubiquitination activity of Ubp1 is required for Hrd1 stabilization. An increase of Hrd1 was also seen when Ubp1 was overexpressed in cells lacking Usa1 or Der1 (*Figure 1B*; lanes 9 versus 4 and lane 10 versus 5).

Cycloheximide-chase experiments confirmed that Hrd1 is rapidly degraded in *hrd3Δ* cells and becomes more stable in cells overexpressing Ubp1 (*Figure 1C*). Overexpression of Ubp1 also inhibited Hrd1 degradation in cells lacking Der1 or Usa1 (*Figure 1—figure supplement 2A,B*). Enzymatically inactive Ubp1 (C110S) had no effect on Hrd1 degradation (*Figure 1C*; *Figure 1—figure supplement 2A,B*). Furthermore, wild-type Ubp1 had no effect when co-expressed with an enzymatically inactive Hrd1 mutant (*Figure 1D*), indicating that Ubp1 reverses autoubiquitination of Hrd1. Consistent with this conclusion, the deletion of Ubp1, which should increase ubiquitination of Hrd1, decreased the levels of Hrd1 (*Figure 1E*; lanes 9–12 versus 1–4). As expected, the additional deletion of the stabilizing Hrd1-partner Hrd3 further reduced Hrd1 levels (*Figure 1E*; lanes 13–16). To further test whether Ubp1 reverses autoubiquitination of Hrd1, we purified His-tagged Hrd1 under denaturing conditions and subjected the material to SDS-PAGE and immunoblotting with anti-ubiquitin antibodies (*Figure 1F*). Indeed, the level of polyubiquitinated Hrd1 was reduced in wild-type or *hrd3Δ* cells overexpressing Ubp1, even though the levels of unmodified Hrd1 were increased (*Figure 1F*) and the total level of ubiquitinated proteins in the cell remained unchanged (*Figure 1—figure supplement 2C*). It is important to note that these samples were not treated with proteasomal inhibitors as in previous experiments (*Carroll and Hampton, 2010*) to prevent the accumulation of ubiquitinated Hrd1 in *hrd3Δ* cells.

To determine whether Hrd1 and Ubp1 interact in vivo, we performed co-immunoprecipitation experiments from yeast cells expressing Hrd1-Flag and Ubp1-V5. Indeed, a small, but reproducible amount of Hrd1-Flag was precipitated with antibodies to the V5 epitope (*Figure 1G*). The low level of co-precipitation may be explained by the fact that Ubp1 is 20-fold more abundant than Hrd1 (*Figure 1—figure supplement 2D*). The specificity of the interaction is supported by control experiments with Orm2, another integral membrane of the endoplasmic reticulum that is approximately 20 times more abundant than Hrd1: Ubp1-V5 did not precipitate Orm2-Flag (*Figure 1G*). The interaction of Hrd1 with Ubp1 was maintained in the absence of Hrd3 or Usa1 (*Figure 1—figure supplement 2E*).

To test whether Ubp1 affects the function of Hrd1 in ERAD, we tested the degradation of the ERAD-L substrate carboxypeptidase Y* (CPY*) (*Finger et al., 1993*) in cycloheximide-chase experiments. In the absence of Ubp1 (*Figure 1H*, solid square with dashed black line), CPY* degradation was slower than in cells expressing Ubp1 (*Figure 1H*, circles with solid black line). Importantly, overexpression of Ubp1 did not prevent degradation of CPY* (*Figure 1H*, triangles with red line) or the steady-state levels of CPY* (*Figure 1—figure supplement 1D*). The degradation of the overexpressed ERAD-L substrate, GFP-CPY* was also promoted by Ubp1 expression (in *Figure 1—figure supplement 3A*). Surprisingly, Ubp1 was not required for ERAD-M and overexpression of Ubp1 only slightly slowed degradation of ERAD-M substrates (*Figure 1—figure supplement 3B and C*). These data support the idea the Ubp1 is promoting optimal ERAD-L, but not ERAD-M. Taken together,

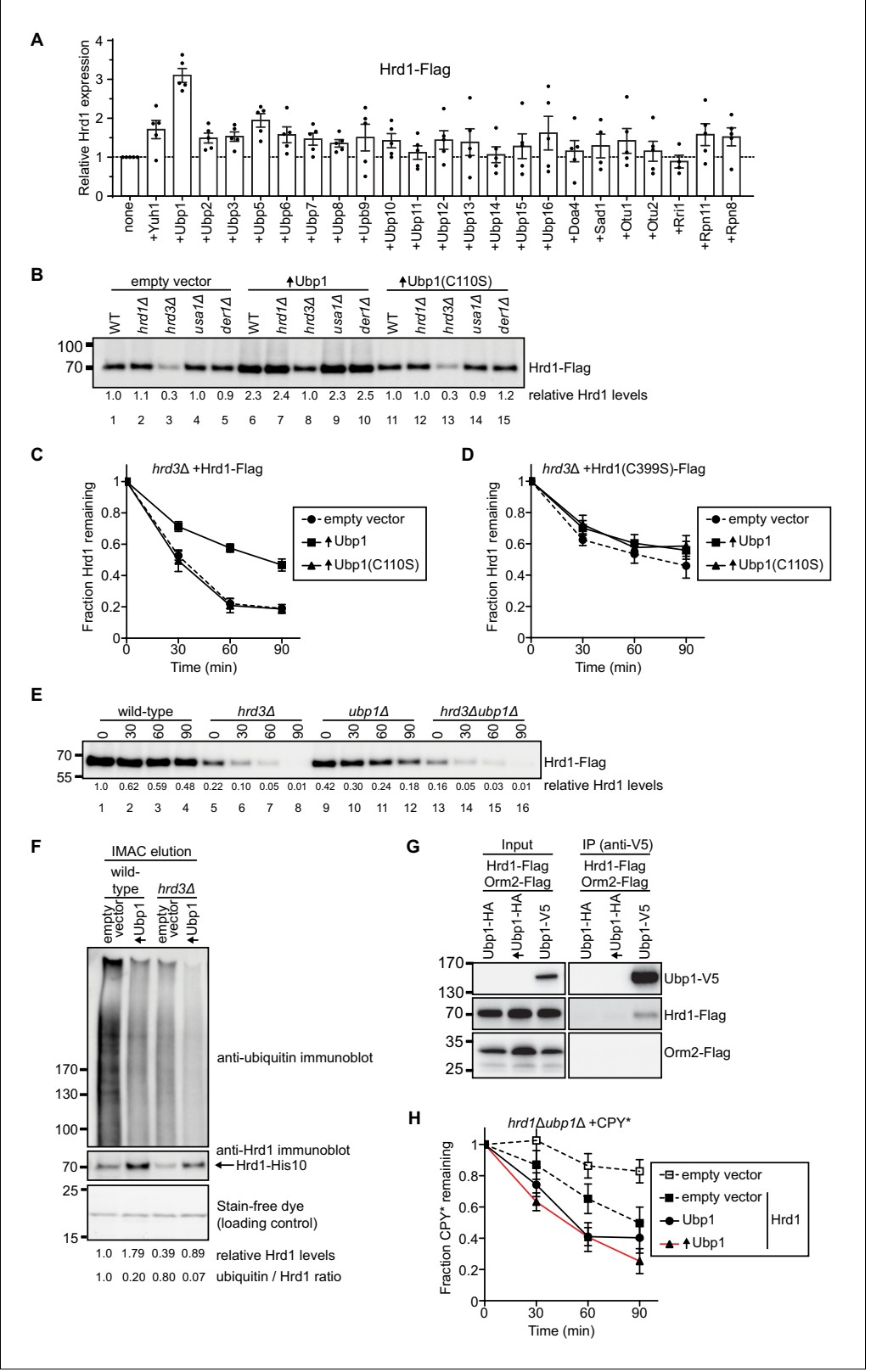

**Figure 1.** Ubp1 stabilizes Hrd1 through deubiquitination. (**A**) Hrd1-Flag was expressed from a centromeric plasmid under its endogenous promoter in *hrd1Δ* cells together with individual deubiquitinating enzymes (DUBs). The DUBs were N-terminally tagged with hemagglutinin (HA) and overexpressed from a centromeric plasmid under the GPD-promoter. Hrd1 levels were analyzed by SDS-PAGE and immunoblotting for the Flag epitope. The data

*Figure 1 continued on next page*

*Figure 1 continued*

are the mean + /- SEM from at least three experiments. (**B**) Comparison of Hrd1-Flag levels in different strains. Where indicated, wild-type Ubp1 or an enzymatically inactive mutant (Ubp1(C110S)) were overexpressed (indicated by upwards-pointing arrows). The numbers below the lanes give quantification of Hrd1-FLAG levels relative to wild-type cells containing an empty vector. This panel is representative of at least three experiments. (**C**) The degradation of Hrd1-Flag was followed in a *hrd3Δ* strain overexpressing the deubiquitinating enzyme Ubp1 or a catalytically-inactive mutant (Ubp1(C110S)). Cycloheximide was added to the cell culture at time point zero, and samples were taken at different time points and analyzed by SDS-PAGE and immunoblotting with anti-Flag antibodies. The data are the mean + /- SEM from at least three experiments. (**D**) As in (**C**) but following the degradation of a catalytically-inactive Hrd1 mutant (Hrd1(C399S)-Flag). (**E**) The degradation of Hrd1-Flag was monitored following addition of cycloheximide in the indicated strains. This panel is representative of at least three experiments. (**F**) Hrd1-His10 was expressed from the endogenous Hrd1 promoter on a centromeric plasmid in either wild-type or *hrd3Δ* cells together with Ubp1 as in (**C**). The membrane proteins were purified with IMAC beads under denaturing conditions, separated by SDS-PAGE, and immunoblotted with anti-Hrd1 and anti-ubiquitin antibodies. The stain-free dye panel shows a non-specific protein band that co-eluted from the IMAC beads and serves to demonstrate equal loading of the samples. This panel is representative of at least three experiments. (**G**) Hrd1-Flag, Orm2-Flag, and Ubp1-V5 were expressed from their endogenous promoters on centromeric plasmids in *ubp1Δ* cells. The lysates were immunoprecipitated with anti-V5 affinity resin, separated by SDS-PAGE and immunoblotted with anti-Flag and anti-V5 antibodies. This panel is representative of at least three experiments. (**H**) The degradation of CPY*-HA was monitored following addition of cycloheximide in the *hrd1Δubp1Δ* strain with Hrd1-Flag and Ubp1-V5 expressed from either its endogenous or GPD promoter on a centromeric plasmid. The data are the mean + /- SEM from at least three experiments. See also *Figure 1—figure supplements 1*, *2* and *3*.

The online version of this article includes the following figure supplement(s) for figure 1:

**Figure supplement 1.** Levels of ERAD components in *S. cerevisiae* cells overexpressing various deubiquitinating enzymes.

**Figure supplement 2.** Hrd1 is stabilized by Ubp1 overexpression.

**Figure supplement 3.** Ubp1 manipulates Hrd1 activity.

these results demonstrate that Hrd1 and Ubp1 interact with one another, and that Ubp1 counteracts the autoubiquitination activity of Hrd1 thereby inhibiting Hrd1 degradation and regulating Hrd1 function.

## Ubp1 function requires its anchoring to the ER membrane

Ubp1 contains an N-terminal transmembrane segment followed by a cytosolic linker and catalytic domain (*Figure 2A*). Ubp1 has been reported to be expressed as two distinct polypeptides; one with the transmembrane segment and the other lacking it (called ΔTM in the scheme in *Figure 2A*), generated from an internal translation start site (Met67) (*Schmitz et al., 2005*). When Ubp1 lacking the transmembrane segment was overexpressed in *hrd3Δ* cells, no Hrd1 stabilization was observed (*Figure 2B*). However, the transmembrane segment of full-length Ubp1 could be replaced by the unrelated transmembrane segment of the Cue4 protein (*Figure 2C*). Even when the entire N-terminal segment preceding the catalytic domain was replaced by a segment derived from Cue4, Ubp1 retained activity in stabilizing Hrd1 (*Figure 2C*). These results demonstrate that the role of the Ubp1 transmembrane segment is primarily to anchor Ubp1 at the ER membrane, which likely facilitates its interaction with the multi-spanning membrane protein Hrd1. In fact, the membrane anchor of Ubp1 could be replaced by the Hrd1-interacting H-domain of Usa1; this soluble fusion protein was able to completely stabilize Hrd1 (*Figure 2C*). The enzymatically inactive version of this construct (H-domain Ubp1 (C110S)) had a small effect on Hrd1 degradation (*Figure 2—figure supplement 1A*), perhaps due to its overexpression. Together, these results suggest that Hrd1 stabilization is primarily achieved by H domain-dependent recruitment of the catalytic domain of Ubp1. Control experiments showed that all Ubp1-constructs were expressed at about the same level (*Figure 2—figure supplement 1B*).

To further test the specificity of Ubp1, we used the cytosolic mammalian DUB Usp2 (*Baker et al., 2005*). Both Ubp1 and Usp2 are members of the ubiquitin-specific protease (USP) family that generally has low ubiquitin-chain linkage specificity (*Mevissen and Komander, 2017*). Overexpression of the catalytic core of Usp2 had only a minor effect on Hrd1 stability (*Figure 2D*). Similar results were

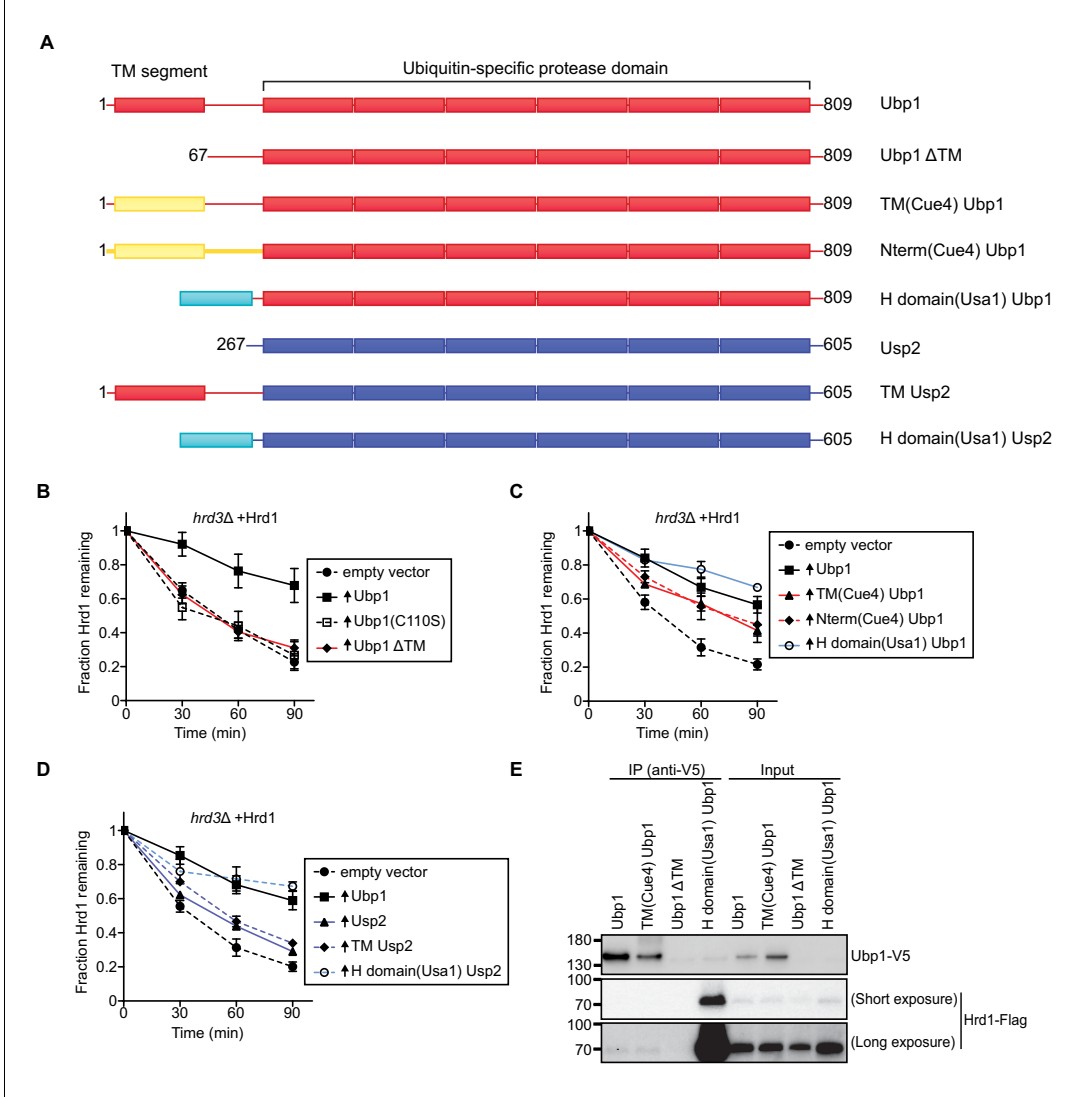

**Figure 2.** Ubp1 function requires its anchoring to the ER membrane. (A) Schematic of chimeric deubiquitinating enzyme constructs used in (B)-(D). TM, transmembrane. Red segments are from Ubp1, yellow ones from Cue4, cyan ones from Usa1, and blue ones from Usp2. The H domain comprises amino acids 437–490 of Usa1. (B) The degradation of Hrd1-Flag was followed in a *hrd3Δ* strain overexpressing the indicated Ubp1 constructs. Cycloheximide was added to the cell culture at time point zero, and samples were taken at the indicated time points and analyzed by SDS-PAGE and immunoblotting with anti-Flag antibodies. The data are the mean + /- SEM from at least three experiments. (C) As in (B), but with other Ubp1 chimeras and wild-type Ubp1 as control. (D) As in (B) with Usp2 chimeras and wild-type Ubp1 as control. (E) Hrd1-Flag and Ubp1-V5 variants were expressed from their endogenous promoters on centromeric plasmids in *hrd1Δubp1Δ* cells. The lysates were immunoprecipitated with anti-V5 affinity resin, separated by SDS-PAGE and immunoblotted with anti-Flag and anti-V5 antibodies. This panel is representative of at least three experiments. See also *Figure 2—figure supplement 1*.

The online version of this article includes the following figure supplement(s) for figure 2:

**Figure supplement 1.** Steady-state levels of overexpressed proteins in *S. cerevisiae* cells.

obtained with a fusion protein containing the transmembrane domain of Ubp1 followed by Usp2 (TM Usp2; *Figure 2D*), indicating that mere membrane targeting of the DUB is insufficient to stabilize Hrd1. However, stabilization was observed when Usp2 was targeted directly to Hrd1 by the Hrd1-interacting H domain of Usa1 (*Figure 2D*), even though this construct was expressed at a lower level than transmembrane Usp2 (*Figure 2—figure supplement 1B*).

Next, we tested if the Ubp1 chimeras co-immunoprecipitated with Hrd1. Both wild-type Ubp1 and TM(Cue4)Ubp1 interacted weakly with Hrd1 (*Figure 2E*). In contrast, Ubp1 missing its transmembrane segment (Ubp1 ΔTM) did not interact with Hrd1 but a soluble version of Ubp1 carrying

the H-domain of Usa1 showed strong interaction with Hrd1 (*Figure 2E*). Thus, there is a general correlation between Ubp1's Hrd1 interaction and its effect on Hrd1 stabilization. Together, these results suggest that wild-type Ubp1 is targeted to Hrd1 by both its transmembrane segment and the USP domain. They also provide further support for the idea that Ubp1 acts on polyubiquitinated Hrd1.

## Ubp1 overexpression bypasses the requirement for Hrd3 in ERAD

Next, we analyzed the effect of Ubp1 overexpression on substrate degradation in yeast strains lacking various ERAD components. As reported previously (*Bordallo et al., 1998*; *Buschhorn et al., 2004*; *Carvalho et al., 2006*; *Denic et al., 2006*; *Knop et al., 1996*), the degradation of CPY* was drastically reduced when either Hrd1, Usa1, Der1, or Hrd3 were absent (compare dashed lines in *Figure 3A* with those in *Figure 3B–E*), confirming that these components are all required for ERAD-L. Ubp1 overexpression only had an effect in *hrd3Δ* cells, where it rescued CPY* degradation (solid line in *Figure 3E* versus those in *Figure 3A–D*). The degradation rate approached that in wild-type cells (*Figure 3A*). These results suggest that Hrd3's major function is to inhibit the ubiquitination activity of Hrd1. Importantly, bypassing Hrd3 by Ubp1 overexpression does not obviate the need for the other ERAD-L components in CPY* degradation (*Figure 3A–D*).

To further test the function of Ubp1, we overexpressed Ubp1 in *hrd1Δhrd3Δ* cells together with wild-type Hrd1. As expected from the results presented before, Ubp1 overexpression stabilized Hrd1 (*Figure 3F*) and allowed degradation of CPY* (*Figure 3G*), even in the absence of Hrd3. A Hrd1 mutant (Hrd1 KRK; Hrd1 with lysine to arginine substitutions at amino acids 319, 325, 366, 368, 370, 371, 373, 387, 407) that cannot be autoubiquitinated in its RING finger domain and is therefore inactive in retrotranslocation in vitro and in vivo (*Baldridge and Rapoport, 2016*), was also stabilized by Ubp1 overexpression (*Figure 3H*). This is consistent with the fact that the protein can still be polyubiquitinated in other domains (*Baldridge and Rapoport, 2016*; *Stein et al., 2014*). However, as expected, the higher Hrd1 KRK levels did not result in CPY* degradation (*Figure 3I*). Ubp1 overexpression had little effect on the stability of catalytically inactive Hrd1(C399S) (*Figure 3J*) or on the degradation of CPY* in Hrd1(C399S) expressing cells (*Figure 3K*), consistent with the notion that Ubp1 acts on polyubiquitinated Hrd1. Because Ubp1 does not accelerate the degradation of CPY* in the presence of retrotranslocation-deficient Hrd1 KRK or inactive Hrd1(C399S) (*Figure 3I and K*), we conclude that Ubp1 overexpression does not bypass normal Hrd1 function.

Stabilization of wild-type Hrd1 by Ubp1 overexpression also resulted in the accelerated degradation of the ERAD-M substrate Erg3 (*Figure 3L*) (*Christiano et al., 2014*; *Jaenicke et al., 2011*). Surprisingly however, the Hrd1 KRK variant was also able to mediate the degradation of Erg3 (*Figure 3M*), in contrast to the ERAD-L substrate CPY* (*Figure 3I*). Overexpression of Ubp1 did not bypass Hrd1 function because inactive Hrd1(C399S) was unable to support the degradation of Erg3 (*Figure 3N*). It therefore appears that autoubiquitination of Hrd1 is not required for the degradation of this ERAD-M substrate.

## The UBL domain of Usa1 inhibits Ubp1 activity

Previous reports demonstrated that Usa1 is required for Hrd1 autoubiquitination and degradation in a *hrd3Δ* strain (*Carroll and Hampton, 2010*; *Vashistha et al., 2016*). This activity could be assigned to the UBL domain of Usa1, but it remained unclear how this domain affects Hrd1 stability (*Carroll and Hampton, 2010*). We wondered if the UBL domain might inhibit Ubp1's DUB activity; in its absence (or absence of the entire Usa1 protein), Ubp1 would effectively reduce ubiquitination and thus degradation of Hrd1.

To test this idea, we used a *hrd3Δusa1Δ* strain, in which Hrd1 is much more stable than in a *hrd3Δ* single deletion strain (*Carroll and Hampton, 2010*; see also *Figure 4—figure supplement 1A*). Nevertheless, there was still some Hrd1 autoubiquitination and degradation in this strain, as overexpression of wild-type Ubp1, but not of enzymatically inactive Ubp1 (C110S), resulted in increased Hrd1 levels (*Figure 4A*, lanes 5–8 versus 9–12). Overexpression of Usa1 lacking its UBL domain (Usa1 ΔUBL) drastically stabilized Hrd1 (lanes 21–24), in contrast to wild-type Usa1, which only had a small effect on Hrd1 stability (lanes 17–20). Thus, the UBL domain of Usa1 indeed seems to attenuate Ubp1 activity.

The co-overexpression of Usa1 and Ubp1 in a *hrd3Δ* strain had little effect on Hrd1 levels and Hrd1 degradation compared to overexpression of Ubp1 alone (*Figure 4B*, lanes 5–8 versus 1–4).

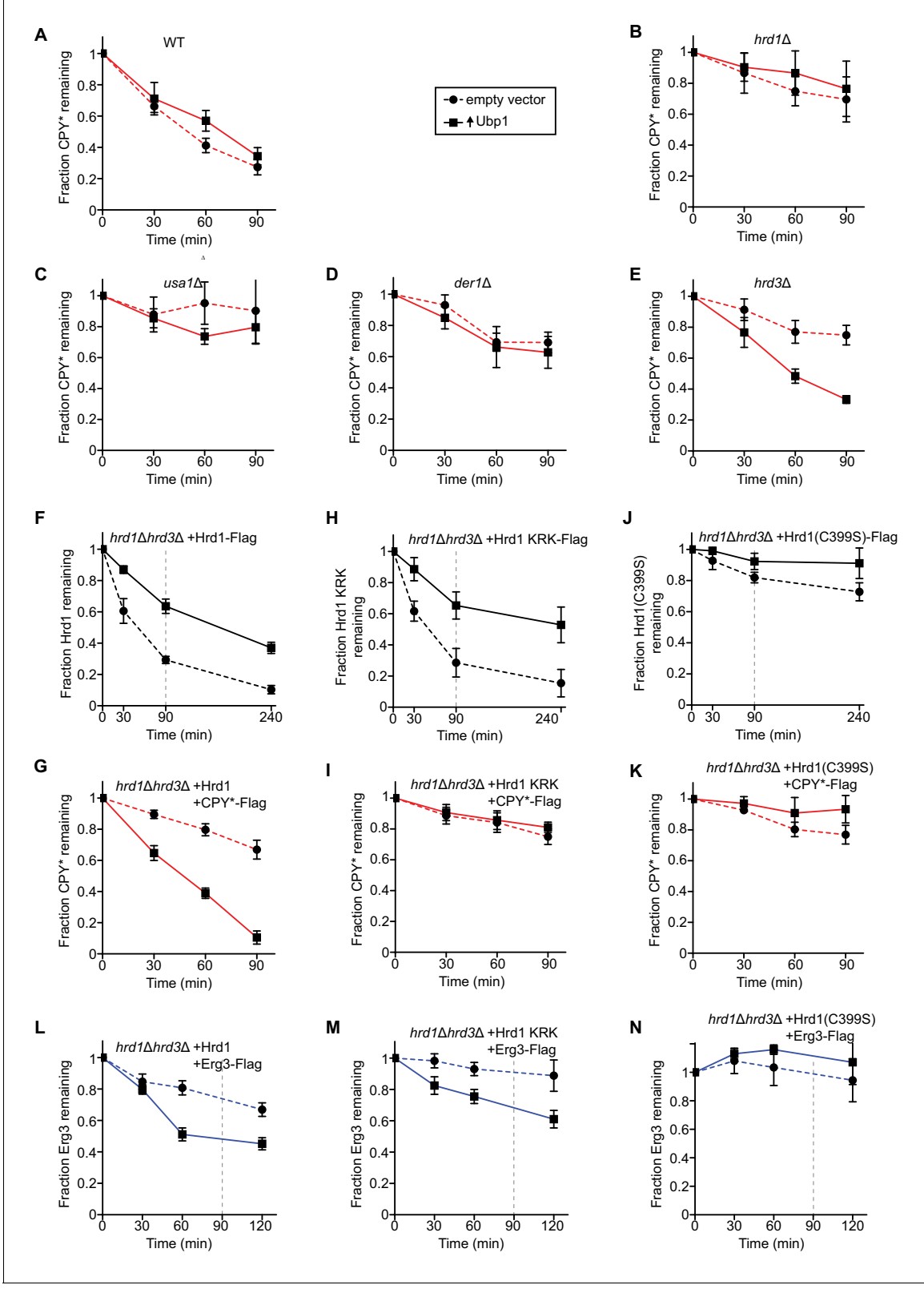

**Figure 3.** Ubp1 overexpression bypasses the requirement for Hrd3 in ERAD. (**A**) The degradation of the ERAD-L substrate CPY*-Flag was tested in wild-type cells with or without overexpression of the deubiquitinating enzyme Ubp1. Cycloheximide was added to the cell culture at time point zero, and samples were taken at the indicated time points and analyzed by SDS-PAGE and immunoblotting with anti-Flag antibodies. The data are the mean + /- SEM from at least three experiments. (**B**) As in (**A**), but with *hrd1Δ* cells. (**C**) As in (**A**), but with *usa1Δ* cells. (**D**) As in (**A**), but with *der1Δ* cells. (**E**) As

*Figure 3 continued on next page*

*Figure 3 continued*

in (A), but with *hrd3Δ* cells. (F) The degradation of Hrd1-Flag was followed by cycloheximide-chase experiments in *hrd1Δhrd3Δ* cells with or without overexpression of the deubiquitinating enzyme Ubp1. The vertical gray dashed line indicates the 90 min end-point in CPY* chase experiments. (G) The degradation of CPY*-Flag was monitored by cycloheximide-chase experiments in *hrd1Δhrd3Δ* cells expressing Hrd1 from its endogenous promoter on a centromeric plasmid together with overexpressed Ubp1. (H) As in (F) but following the degradation of a Hrd1 variant that cannot be autoubiquitinated in its RING finger domain (Hrd1 KRK-Flag). (I) As in (G), but with Hrd1 KRK. (J) As in (F), but following the degradation of catalytically-inactive Hrd1 (Hrd1(C399S)-Flag). (K) As in (G), but with Hrd1(C399S). (L) The degradation of the ERAD-M substrate Erg3-Flag was monitored by cycloheximide-chase experiments in *hrd1Δhrd3Δ* cells expressing Hrd1 from its endogenous promoter on a centromeric plasmid together with overexpressed Ubp1. The vertical gray dashed line indicates the 90 min end-point used for CPY* chases. (M) As in (L), but with Hrd1 KRK. (N) As in (L), but with Hrd1(C399S).

However, when Usa1 ΔUBL was co-expressed with Ubp1, Hrd1 was strongly stabilized (lanes 9–12), indicating that the UBL domain of Usa1 indeed inhibits Ubp1 activity. In a triple-deletion mutant (*hrd3Δubp1Δusa1Δ*), Hrd1 was rapidly degraded (*Figure 4C*, lanes 1–4), but could be stabilized by its partner protein Hrd3 (lanes 5–8) or by increasing deubiquitination through Ubp1 overexpression (lanes 9–12). However, as expected from our model, Usa1 or Usa1 ΔUBL had no effect in the absence of Ubp1 (lanes 17–20 and 21–24). When Ubp1 was expressed from its endogenous promoter in the triple-deletion mutant (*hrd3Δubp1Δusa1Δ*), the additional expression of Usa1 ΔUBL caused stabilization of Hrd1 (*Figure 4D*, lanes 9–12), whereas expression of wild-type Usa1 only had a small effect (*Figure 4D*, lanes 5–8). Collectively, these results support the idea that the UBL domain of Usa1 inhibits the activity of Ubp1.

To test whether Hrd1 activity was affected by Usa1 or Usa1 ΔUBL, we followed the degradation of both ERAD-L and ERAD-M substrates. Overexpression of Usa1 partially inhibited CPY* turnover (*Figure 4—figure supplement 1B*, lanes 9–12 and *Carvalho et al., 2010*) and overexpression of Usa1 ΔUBL had a slightly stronger effect (*Figure 4—figure supplement 1B*, lanes 13–16). To test whether this effect was limited to ERAD-L substrates, we followed the degradation of ERAD-M substrates. We tested the constitutive degradation of the Hmg2 non-responder mutant (Hmg2-NR1-GFP; substitution of amino acids 348–352 TFYSA to ILQAS) and also the regulated degradation of Hmg2-GFP by flow cytometry (*Shearer and Hampton, 2005*). Degradation of these ERAD-M substrate still required Hrd1 but was unaffected by the overexpressing Usa1 or Usa1 ΔUBL (*Figure 4—figure supplement 1C and D*). The fact that overexpression of Usa1 and Usa1 ΔUBL disrupted ERAD-L, but not ERAD-M, suggests that ERAD-L was likely compromised by overexpressed Usa1 sequestering Der1 (*Carvalho et al., 2010*; *Horn et al., 2009*). Taken together, these results support the idea that the UBL domain does not affect substrate degradation directly (*Carroll and Hampton, 2010*; *Vashistha et al., 2016*).

## Discussion

Our results reveal a novel mechanism by which the central ERAD component Hrd1 is regulated (*Figure 4E*). Previous experiments had shown that Hrd1 is activated for ERAD by autoubiquitination; polyubiquitinated Hrd1 allowed a polypeptide segment of the substrate to move across the membrane (*Baldridge and Rapoport, 2016*; *Stein et al., 2014*). This mechanism, however, raised the question of how polyubiquitinated Hrd1 escapes degradation and returns to its inactive ground state. Our results now indicate that Ubp1, a membrane-bound DUB, is responsible for resetting Hrd1 (*Figure 4E*). Ubp1 counteracts Hrd1's autoubiquitination and therefore reduces its degradation (*Figure 1B–F*), while also regulating Hrd1's activity in ERAD-L (*Figure 1H* and *Figure 1—figure supplement 3A–C*). Ubp1 is an unusual DUB because it has an N-terminal transmembrane segment that is required for its activity towards Hrd1. However, there seem to be additional, so far unidentified, determinants of specificity within the catalytic domain of Ubp1 (*Figure 2*). The activity of Ubp1 must be tightly regulated, as excessive deubiquitination would prevent Hrd1 autoubiquitination and activation, and inadequate deubiquitination would result in Hrd1 degradation. Our results show that Usa1, a component associated with Hrd1, regulates Ubp1 activity; Usa1's UBL domain attenuates the activity of Ubp1 and thereby allows autoubiquitination of Hrd1 (*Figure 4E*). Autoubiquitination is also regulated by Hrd3, a lumenal binding partner of Hrd1 (*Gardner et al., 2000*). Hrd3 serves as a brake of Hrd1's ubiquitination activity (*Figure 4E*). We propose that substrate binding to Hrd3

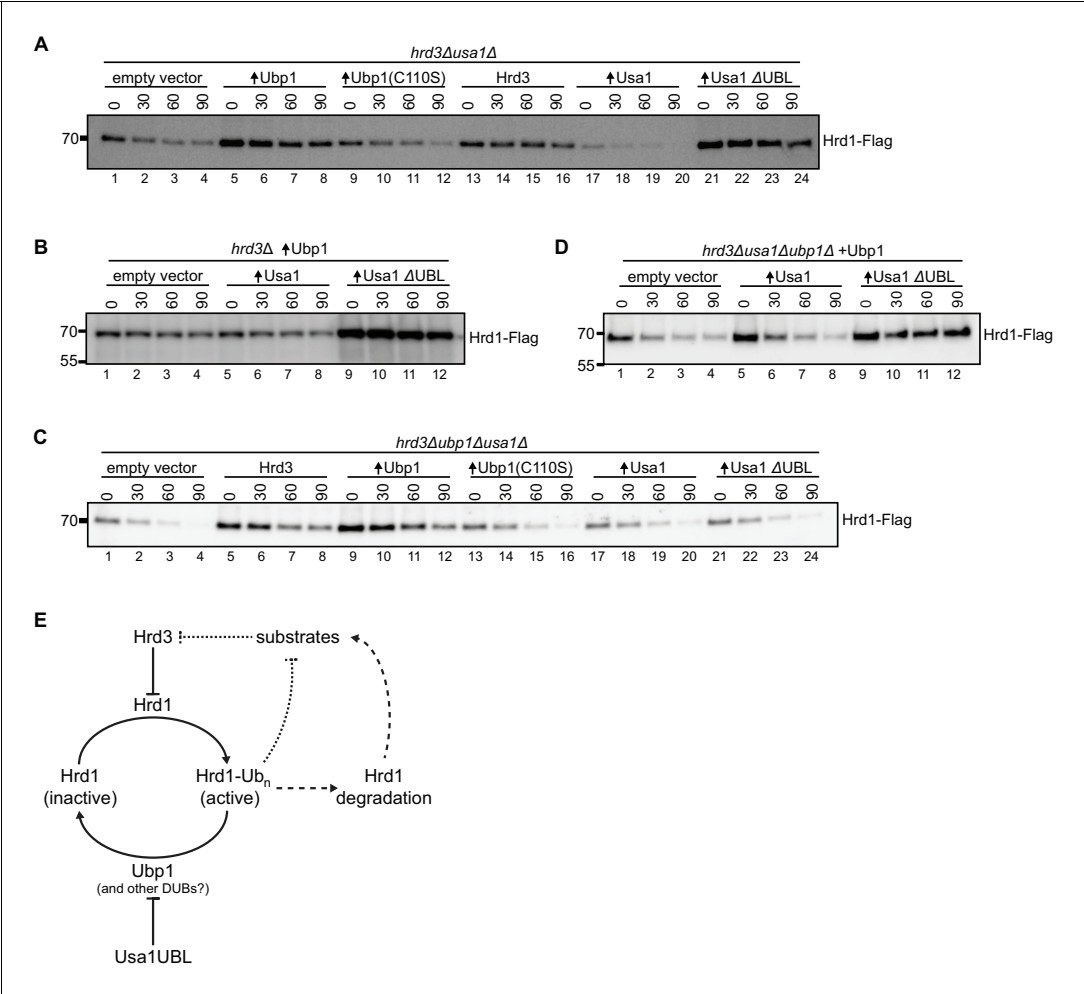

**Figure 4.** The UBL domain of Usa1 inhibits Ubp1 activity. (**A**) Degradation of Hrd1-Flag in *hrd3Δusa1Δ* cells overexpressing the indicated proteins analyzed by cycloheximide-chase experiments. Ubp1(C110) is an enzymatically-inactive version of Ubp1, and Usa1ΔUBL is Usa1 lacking its UBL domain. This panel is representative of at least three experiments. (**B**) Degradation of Hrd1-Flag in *hrd3Δ* cells overexpressing both Ubp1 and wild-type Usa1 or Usa1ΔUBL. This panel is representative of at least three experiments. (**C**) Degradation of Hrd1-Flag in *hrd3Δubp1Δusa1Δ* cells overexpressing the indicated proteins. This panel is representative of at least three experiments. (**D**) Degradation of Hrd1-Flag in *hrd3Δubp1Δusa1Δ* cells expressing Ubp1 and overexpressing the indicated proteins. This panel is representative of at least three experiments. (**E**) Model for the regulation of Hrd1 by autoubiquitination and deubiquitination. Hrd3 serves as an inhibitor of Hrd1 autoubiquitination. Binding of an ERAD substrate to Hrd3 would relieve this inhibition, resulting in Hrd1 activation and degradation of the substrate (dotted line). Hrd1 ubiquitination also leads to Hrd1 degradation, which counteracts substrate degradation (broken lines). Ubp1 reverses autoubiquitination of Hrd1 and returns Hrd1 to the inactive state. Ubp1 activity is attenuated by the UBL domain of the Hrd1-associated component Usa1. See also *Figure 4—figure supplement 1*.

The online version of this article includes the following figure supplement(s) for figure 4:

**Figure supplement 1.** The Usa1 UBL domain influences Hrd1 stability.

releases this brake (*Figure 4E*), allowing autoubiquitination of Hrd1, which in turn opens the channel for retrotranslocation of the substrate. Hrd3 is not absolutely required for a basic ERAD process, as substrates are still degraded in its absence when Hrd1 is stabilized by Ubp1 overexpression. Taken together, our results indicate that Hrd1 undergoes cycles of autoubiquitination and deubiquitination, which are regulated by Hrd1-interacting components.

Surprisingly, we found that the ERAD-M substrate Erg3 does not require Hrd1 autoubiquitination or deubiquitination for its degradation. It is therefore possible that the model depicted in *Figure 4E* is only applicable to ERAD-L substrates. ERAD-M substrates might enter Hrd1 laterally with their transmembrane segments and perhaps this does not require autoubiquitination of Hrd1. Alternatively, and more likely, ERAD-M (and ERAD-C) substrates may use the Der1 homolog Dfm1 for their

extraction into the cytosol (*Neal et al., 2018*). Dfm1 seems to mediate the cytosolic extraction of these ERAD substrates even in the absence of Hrd1 (or Doa10), which would explain why autoubiquitination and deubiquitination of Hrd1 is not required.

The presence of Ubp1 is required for optimal performance of ERAD-L (*Figure 1H*). The decreased degradation rate in the absence of Ubp1 can be explained by decreased Hrd1 activity. Or alternately, decreased ERAD-L degradation could be explained by enhanced Hrd1 degradation caused by its insufficient deubiquitination. The effect of Ubp1 deletion is relatively small, perhaps because a combination of other DUBs can replace Ubp1. The overexpression of Ubp1 has no effect on the degradation of ERAD-L substrates and slightly slows the degradation of ERAD-M substrates (*Figure 1—figure supplement 3B,C* and *Figure 4—figure supplement 1C,D*). Perhaps, ERAD-M substrates are affected because they are accessible to Ubp1 on the cytosolic side of the membrane prior to their extraction from the membrane by Dfm1 (*Neal et al., 2018*). An alternate possibility is that deubiquitination of Hrd1 reduces its activity towards ERAD-M substrates. While Ubp1 does slow down the degradation rate of ERAD-M substrates, this balance could be advantageous for substrate specificity (*Zhang et al., 2013*).

Our results explain several puzzling observations in the literature. It has long been known that the deletion of Hrd3, which interacts with Hrd1 on the ER lumenal side, results in excessive Hrd1 autoubiquitination and degradation (*Gardner et al., 2000*). These reactions were attenuated when Usa1 or Usa1's UBL domain were also deleted (*Carroll and Hampton, 2010*). Our results now explain these results by the demonstration that the UBL domain inhibits Ubp1. In the absence of this domain, Ubp1 is hyperactive and reduces polyubiquitination of Hrd1 and thus its subsequent degradation (*Figure 4E*). This model also explains why the UBL domain of Usa1 has no direct effect on the degradation of ERAD substrates (*Figure 4—figure supplement 1B–D*) (*Carroll and Hampton, 2010*). Whether the UBL domain of Usa1 inhibits Ubp1 by a physical interaction or indirectly remains to be determined. However, functional interactions between UBL domains and DUBs have been reported, with DUBs being allosterically regulated or competitively inhibited by specific UBL domains (*Faesen et al., 2012*).

Although the absence of the UBL domain of Usa1 restored wild-type levels of Hrd1 in *hrd3∆* cells, substrate degradation was still defective (*Vashistha et al., 2016*). Our experiments show that both Hrd1 levels and ERAD can be restored in a *hrd3∆* strain when Ubp1 is overexpressed. Thus, while Hrd3 is normally involved in ERAD, we suggest it primarily performs a supporting role for Hrd1 through regulating Hrd1's autoubiquitination activity and helping to recruit ERAD substrates (*Carvalho et al., 2010*; *Gardner et al., 2001*). Clearly, Hrd1 regulation is complex and one explanation for the previously published results is that Hrd1 autoubiquitination was reduced to a level at which channel opening was severely inhibited. In contrast, Ubp1 overexpression (in the presence of Usa1) might still allow a low level of autoubiquitination that is sufficient for retrotranslocation, but not for efficient Hrd1 degradation. This interpretation implies that Ubp1 overexpression does not completely overcome the inhibitory function of the UBL domain, perhaps because the interaction between Hrd1 and Ubp1 is weak (*Figure 1G*) while that between Hrd1 and Usa1 is strong, or because Usa1 sterically prevents the access of Ubp1 to Hrd1 (*Carvalho et al., 2006*; *Horn et al., 2009*).

At this point, it is unclear exactly how Ubp1 is recruited to Hrd1. Deletion of either Hrd3 or Usa1 doesn't have a strong effect on the Hrd1/Ubp1 interaction (*Figure 1—figure supplement 2E*). This could mean Hrd1 and Ubp1 interact directly, or that Ubp1 is recruited through the Hrd1-attached ubiquitin chain. Another possibility is that the interaction is mediated by another, unidentified protein, which might explain the low level of co-precipitation of Ubp1 and Hrd1 (*Figure 1G* and *Figure 1—figure supplement 2E*).

While Hrd1 polyubiquitinates both itself and ERAD substrates, deubiquitination of these components occurs by separate mechanisms. It is somewhat surprising that CPY* degradation is not accelerated by the increased Hrd1 levels caused by Ubp1 overexpression (*Figures 1H* and *3A*). However, Hrd1 functions in a complex with other components, including Hrd3, Usa1, and Der1, the levels of which remain unchanged (*Figure 1—figure supplement 1*). Importantly, overexpression of Ubp1 does not prevent the degradation of CPY* (*Figures 1H* and *3A*), indicating that Ubp1 does not act on polyubiquitinated ERAD-L substrate. In fact, an ERAD substrate needs to retain its polyubiquitin chain until it has been extracted from the membrane by the Cdc48/p97 ATPase complex, because the polyubiquitin chain serves as a recognition signal for the Ufd1/Npl4 cofactor of Cdc48. Substrate

deubiquitination occurs by cytosolic DUBs, one of which is Otu1 (Yod1 in mammals), an enzyme that binds to the Cdc48 ATPase through its UBX-like domain (*Stein et al., 2014*). As long as the ATPase complex is bound to the membrane, Cdc48 interacts with Ubx2, which prevents access of Otu1. Once the ATPase has pulled the substrate out of the membrane and has moved into the cytosol, Otu1 can trim the ubiquitin chain, the substrate is released from Cdc48/p97, and then transferred to the proteasome (*Bodnar and Rapoport, 2017*; *Ernst et al., 2009*; *Neal et al., 2017*). Thus, ERAD involves two distinct deubiquitination reactions, one catalyzed by Ubp1 at the ER membrane to reset Hrd1, and another catalyzed by Otu1 in the cytosol to release substrate from Cdc48/p97.

Integral membrane DUBs are relatively rare. In both yeast and mammals, there are only two DUBs that are integral membrane proteins (out of 24 and 90 enzymes, respectively) (*Komander et al., 2009*). In *S. cerevisiae*, Ubp1 is the only known DUB in the ER membrane (*Schmitz et al., 2005*). Our results show that the N-terminal membrane anchor is required for reversing the ubiquitin-modification of Hrd1 but that it can be replaced by a transmembrane segment from a different protein. The apparent 20-fold excess of Ubp1 over Hrd1 makes it is likely that Ubp1 has additional substrates which could contribute indirectly to Hrd1 stability. A shorter version of Ubp1, which lacks the transmembrane segment, has been reported to have a role in endocytosis (*Schmitz et al., 2005*).

Although one would expect the function of Ubp1 to be evolutionarily conserved, there is no obvious homolog in higher organisms. However, Usp19 has been reported to serve as a DUB for polyubiquitinated Hrd1 (*Harada et al., 2016*). Usp19 is present at the ER membrane (*Hassink et al., 2009*), is proposed to play a role in ERAD (*Lee et al., 2014*), and is one of two mammalian DUBs with transmembrane segments (*Komander et al., 2009*). Mammalian Hrd1 is associated with HERP (*Schulze et al., 2005*), a putative Usa1 homolog with a UBL domain (*Kokame et al., 2000*), but whether Usp19 functions in a similar way as Ubp1 in yeast remains to be investigated.

# Materials and methods

## Key resources table

| Reagent type (species) or resource | Designation | Source or reference | Identifiers | Additional information |
|---|---|---|---|---|
| Gene (*Saccharomyces cerevisiae*) | HRD1 | This study | YOL013C | Amplified from BY4741 (genomic DNA) |
| Gene (*S. cerevisiae*) | HRD3 | This study | YLR207W | BY4741 (genomic DNA) |
| Gene (*S. cerevisiae*) | USA1 | This study | YML029W | BY4741 (genomic DNA) |
| Gene (*S. cerevisiae*) | DER1 | This study | YBR201W | BY4741 (genomic DNA) |
| Gene (*S. cerevisiae*) | YUH1 | This study | YJR099W | BY4741 (genomic DNA) |
| Gene (*S. cerevisiae*) | UBP1 | This study | YDL122W | BY4741 (genomic DNA) |
| Gene (*S. cerevisiae*) | UBP2 | This study | YOR124C | BY4741 (genomic DNA) |
| Gene (*S. cerevisiae*) | UBP3 | This study | YER151C | BY4741 (genomic DNA) |
| Gene (*S. cerevisiae*) | UBP5 | This study | YER144C | BY4741 (genomic DNA) |
| Gene (*S. cerevisiae*) | UBP6 | This study | YFR010W | BY4741 (genomic DNA) |
| Gene (*S. cerevisiae*) | UBP7 | This study | YIL156W | BY4741 (genomic DNA) |
| Gene (*S. cerevisiae*) | UBP8 | This study | YMR223W | BY4741 (genomic DNA) |
| Gene (*S. cerevisiae*) | UBP9 | This study | YER098W | BY4741 (genomic DNA) |
| Gene (*S. cerevisiae*) | UBP10 | This study | YNL186W | BY4741 (genomic DNA) |
| Gene (*S. cerevisiae*) | UBP11 | This study | YKR098C | BY4741 (genomic DNA) |
| Gene (*S. cerevisiae*) | UBP12 | This study | YJL197W | BY4741 (genomic DNA) |
| Gene (*S. cerevisiae*) | UBP13 | This study | YBL067C | BY4741 (genomic DNA) |
| Gene (*S. cerevisiae*) | UBP14 | This study | YBR058C | BY4741 (genomic DNA) |
| Gene (*S. cerevisiae*) | UBP15 | This study | YMR304W | BY4741 (genomic DNA) |
| Gene (*S. cerevisiae*) | UBP16 | This study | YPL072W | BY4741 (genomic DNA) |

*Continued on next page*

Continued

| Reagent type (species) or resource | Designation | Source or reference | Identifiers | Additional information |
|---|---|---|---|---|
| Gene (*S. cerevisiae*) | *DOA4* | This study | YDR069C | BY4741 (genomic DNA) |
| Gene (*S. cerevisiae*) | *SAD1* | This study | YFR005C | BY4741 (genomic DNA) |
| Gene (*S. cerevisiae*) | *OTU1* | This study | YFL044C | BY4741 (genomic DNA) |
| Gene (*S. cerevisiae*) | *OTU2* | This study | YHL013C | BY4741 (genomic DNA) |
| Gene (*S. cerevisiae*) | *RRI1* | This study | YDL216C | BY4741 (genomic DNA) |
| Gene (*S. cerevisiae*) | *RPN11* | This study | YFR004W | BY4741 (genomic DNA) |
| Gene (*S. cerevisiae*) | *RPN8* | This study | YOR261C | BY4741 (genomic DNA) |
| Gene (*S. cerevisiae*) | *ORM2* | This study | YLR350W | BY4741 (genomic DNA) |
| Gene (*S. cerevisiae*) | *ERG3* | This study | YLR056W | BY4741 (genomic DNA) |
| Gene (*S. cerevisiae*) | *HMG2* | This study | YLR450W | BY4741 (genomic DNA) |
| Gene (*S. cerevisiae*) | *PRC1* (CPY) | This study | YMR297W | BY4741 (genomic DNA) |
| Gene (*Homo sapiens*) | Ubiquitin specific protease 2; Usp2 | This study | AAP36388.1 | Synthetic gene |
| Strain, strain background (*S. cerevisiae*) | BY4741 | GE Dharmacon | | MATa *his3Δ1 leu2Δ0 met15Δ0 ura3Δ0* |
| Strain, strain background (*S. cerevisiae*) | *hrd1Δ* | GE Dharmacon | | MATa *his3Δ1 leu2Δ0 met15Δ0 ura3Δ0 hrd1::kanR* |
| Strain, strain background (*S. cerevisiae*) | *hrd3Δ* | GE Dharmacon | | MATa *his3Δ1 leu2Δ0 met15Δ0 ura3Δ0 hrd3::kanR* |
| Strain, strain background (*S. cerevisiae*) | *usa1Δ* | GE Dharmacon | | MATa *his3Δ1 leu2Δ0 met15Δ0 ura3Δ0 usa1::kanR* |
| Strain, strain background (*S. cerevisiae*) | *der1Δ* | GE Dharmacon | | MATa *his3Δ1 leu2Δ0 met15Δ0 ura3Δ0 der1::kanR* |
| Strain, strain background (*S. cerevisiae*) | *ubp1Δ* | This study | | MATa *his3Δ1 leu2Δ0 met15Δ0 ura3Δ0 ubp1::kanR* |
| Strain, strain background (*S. cerevisiae*) | yRB057A | This study | | MATa *his3Δ1 leu2Δ0 MET15 lys2Δ0 ura3Δ0 hrd1::kanR hrd3::kanR* |
| Strain, strain background (*S. cerevisiae*) | yBGP15A | This study | | MATa *his3Δ1 leu2Δ0 met15Δ0 ura3Δ0 hrd1::kanR ubp1::kanR* |
| Strain, strain background (*S. cerevisiae*) | yRB0065A | This study | | MATa *his3Δ1 leu2Δ0 met15Δ0 ura3Δ0 hrd3::kanR usa1::kanR* |
| Strain, strain background (*S. cerevisiae*) | yRB0126 | This study | | MATa *his3Δ1 leu2Δ0 met15Δ0 ura3Δ0 hrd3::kanR ubp1::hphNT1* |
| Strain, strain background (*S. cerevisiae*) | yRB0075C | This study | | MAT? *his3Δ1 leu2Δ0 MET15 ura3Δ0 hrd3::kanR usa1::kanR ubp1::hphNT1* |
| Genetic reagent (*S. cerevisiae*) | CPY* | (*Finger et al., 1993*) | | G255R point mutation in *PRC1* (CPY) leading to CPY* |
| Genetic reagent (*S. cerevisiae*) | GPD promoter | This study | YGR192C | 667 basepairs upstream of *TDH3* (GPD) gene; BY4741 (genomic DNA) |

*Continued*

| Reagent type (species) or resource | Designation | Source or reference | Identifiers | Additional information |
|---|---|---|---|---|
| Antibody | THE Anti-DYKDDDK Antibody (mAb, mouse) | Genscript | A00187 | 1:2000 (5% milk in TBST) |
| Antibody | Anti-HA High Affinity Antibody (mAb, clone 3F10, rat) | Roche | 11867423001 | 1:2000 (5% milk in TBST) |
| Antibody | THE V5 Tag Antibody (mAb, mouse) | Genscript | A01724 | 1:2500 (5% milk in TBST) |
| Antibody | Anti-PGK1 (mAb, clone 22C5D8, mouse) | Invitrogen | 459250 | 1:1000 (5% milk in TBST) |
| Antibody | Anti-Ubiquitin (mAb, clone P4D1, mouse) | Santa Cruz Biotechnologies | sc-8017 | 1:500 (5% milk in TBST) |
| Antibody | Amersham ECL Rabbit IgG, HRP-linked whole Ab (donkey) | GE Healthcare | NA934 | 1:4000 (5% milk in TBST) |
| Antibody | Amersham ECL Mouse IgG, HRP-linked whole Ab (sheep) | GE Healthcare | NA931 | 1:4000 (5% milk in TBST) |
| Antibody | Amersham ECL Rat IgG, HRP-linked whole antibody (goat) | GE Healthcare | NA935 | 1:4000 (5% milk in TBST) |
| Commercial assay or kit | Western Lightning Plus-ECL, Enhanced Chemiluminescence Substrate | Perkin-Elmer | NEL103E001 | |
| Commercial assay or kit | Amersham ECL Select Western Blotting Detection Reagent | GE Healthcare | RPN2235 | |
| Chemical compound, drug | SYTOX Blue Nucleic Acid Stain | Invitrogen | S11348 | |
| Chemical compound, drug | Cycloheximide | Calbiochem | 239763 | |
| Chemical compound, drug | Decyl Maltose Neopentyl Glycol (DMNG) | Anatrace | NG322 | |
| Chemical compound, drug | Zaragozic Acid A | Cayman Chemical | 17452 | |
| Software, algorithm | Graphpad Prism 8 | Graphpad Software LLC. | | version 8.3.0 |
| Software, algorithm | FlowJo | Becton, Dickinson and Company | | Version 10.6.0 |
| Other | Dynabeads His-Tag Isolation and Pulldown | Invitrogen | 10104D | |
| Other | Anti-V5 Agarose Affinity Gel | Millipore | A7345 | |

## Yeast strains and plasmids

Deletion strains used in this study were purchased from GE Dharmacon and are derivatives of BY4741 (MATa *his3Δ1 leu2Δ0 met15Δ0 ura3Δ0*) or BY4742 (MATα *his3Δ1 leu2Δ0 lys2Δ0 ura3Δ0*) (see *Supplementary file 1*). The *ubp1Δ* strain was generated by a transforming a PCR-amplified targeting cassette with the LiAc/PEG methods (*Gietz and Schiestl, 2007*). Double and triple deletion strains were generated by crossing and sporulation. Genotypes were verified by PCR. Plasmids were constructed using standard restriction cloning or Gibson assembly. All plasmids used in this study were

centromeric plasmids (*Sikorski and Hieter, 1989*) and where indicated, overexpression was driven by the GPD/*TDH3* promoter. For a list of plasmids used in this study, see *Supplementary file 2*.

## Degradation assays

Cycloheximide-chase degradation assays were performed as described previously (*Gardner et al., 1998*) with the following modifications. Cells were grown to mid-log phase (0.4–0.7 $OD_{600}$/mL) in synthetic dropout media. The cells were pelleted at 2000 x g for 5 min and resuspended to 2.5 $OD_{600}$/mL in fresh media. At time '0 min' the culture was supplemented with 50 µg/mL cycloheximide and an aliquot was taken and centrifuged as above. The cell pellet was either flash frozen in liquid $N_2$ or resuspended in lysis buffer. The remaining culture was incubated at 30˚C with samples taken as indicated.

Cell pellets were resuspended in lysis buffer (10 mM MOPS, pH 6.8, 1% SDS, 8M urea, 10 mM EDTA, fresh protease inhibitors) at 25 $OD_{600}$/mL with an equivalent volume of acid-washed glass beads (0.1 mm, Bio-Spec). After vortexing for 2 min, an equal volume of urea sample buffer (125 mM Tris pH 6.8, 4% SDS, 8M urea, 10% β-mercaptoethanol) was added and mixed. The samples were incubated at 65˚C for 5 min before SDS-PAGE, transfer to PVDF membrane and immunoblotting (THE Anti-DYKDDDK Antibody, Genscript; HA clone 3F10, Roche) and detection by chemiluminescence with Western Lightning Plus-ECL (Perkin-Elmer) on a ChemiDoc MP (Bio-Rad). For quantification of the immunoblot band intensities, band intensities were normalized based upon total protein in the sample quantified within the same gels using Bio-Rad Stain-Free Dye Imaging Technology or by normalization to the intensity of PGK1 (clone 22C5D8, Thermo Scientific).

Cells expressing GPD-driven Hmg2-GFP or Hmg2-NR1-GFP with the indicated plasmids were grown to mid-log phase in synthetic dropout media. Cells were pelleted and resuspended in synthetic medium and transferred into a 96-well round bottom plate. To follow degradation of Hmg2-NR1-GFP, the medium was supplemented with 50 µg/mL cycloheximide. To induce degradation of Hmg2-GFP, the media was supplemented with 10 µg/mL zaragozic acid (Cayman Chemical). The cells were grown for 4 hr, then pelleted and washed with ice-cold PBS (137 mM NaCl, 2.7 mM KCl, 10 mM $Na_2HPO_4$, 1.8 mM $KH_2PO_4$, pH 7.4). The washed cells were resuspended at 1 $OD_{600}$/mL in PBS containing the viability dye Sytox Blue (Invitrogen) at 1 µM. Cells were kept at 4˚C prior to flow cytometry using a Bio-Rad ZE5 Cell Analyzer with Everest software. At least ten thousand events were analyzed using forward/side scatter to identify single cells and Sytox Blue fluorescence was used to exclude dead cells. GFP fluorescence was measured from the 488 nm laser with a 509 nm/24 nm bandpass filter set, while Sytox Blue fluorescence was measured from the 405 nm laser with a 460 nm/22 nm bandpass filter set. Data were analyzed and figures were generated using FlowJo V10.6 (FlowJo LLC.). The low-expressing GFP population is present in all traces and is indicated under the dashed lines.

## Detection of Hrd1 ubiquitination in vivo

Denaturing pulldowns of Hrd1 were performed as described previously (*Baldridge and Rapoport, 2016*) with the following modifications. Cells with a centromeric plasmid bearing an endogenous Hrd1 promoter and a Hrd1-His10 were grown to mid-log phase and lysed in 50 mM HEPES pH 7.4, 300 mM KCl, protease inhibitors, 1 mM PMSF, 1.5 µM pepstatin, 8M urea (to prevent additional Hrd1 autoubiquitination), and 5 mM NEM (to inhibit deubiquitinating enzymes). The lysates were centrifuged at 2000 x g for 10 min, and the supernatant was collected and re-centrifuged for 30 min in a Ti45 rotor at 42,000 rpm ($RCF_{avg}$ 138,001). The membranes were solubilized in 50 mM HEPES pH 7.4, 300 mM KCl, protease inhibitors, 1 mM PMSF, 6M urea, 1.5% Triton X-100 final and 25 mM imidazole) for 1 hr at 4˚C. His-tag Dynabeads (Life Technologies) were added (0.25 mL per 1,500 OD cells) and incubated for an additional 1 hr. The beads were washed three times with a 30-fold excess buffer. Hrd1-His10 was eluted with buffer including 400 mM imidazole. The samples were analyzed by SDS-PAGE and immunoblotting with anti-Hrd1 and anti-ubiquitin antibodies (clone P4D1, Santa Cruz). Unbound IMAC flow-through was used for a loading control to demonstrate equal material input.

## Immunoprecipitation of Hrd1 and interacting partners

Cells with a centromeric plasmids bearing a combination of Hrd1-3xFlag, Orm2-3xFlag, and endogenous or GPD driven Ubp1-3xHA,Ubp1-3xV5, or Ubp1-3xV5 variants were grown to mid-log phase, pelleted, washed once with water and flash frozen in liquid nitrogen. Cells were thawed on ice and treated for 10 min with spheroplasting buffer (50 mM HEPES, pH 7.4, 150 mM NaCl, 1M sorbitol) supplemented with 10 mM DTT. Cells were pelleted and washed once with spheroplasting buffer before resuspending in spheroplasting buffer supplemented with 20 µg zymolyase 100T (per 5 OD). Cells were incubated at 30°C for 60 min, then pelleted at 3200 x g for 5 min. The spheroplasts were resuspended in IP buffer (50 mM HEPES, pH 7.4, 150 mM NaCl with freshly added 1 mM PMSF and 3 µM pepstatin A) and lysed with 7–10 strokes of a tight-fitting Dounce homogenizer. Lysed cells were pelleted at 20,000 x g for 20 min. The supernatant was discarded and the pellet (microsome fraction) was resuspended in IP buffer supplemented with 1% Decyl Maltose Neopentyl Glycol (DMNG). Microsomes were solubilized for 1 hr at 4°C, before input samples were taken and 7.5 ODs of solubilized proteins were mixed with 20 µL (40 µL slurry) Anti-V5 Agarose Affinity Gel (Millipore, A7345) and rolled for 2 hr at 4°C. The bound proteins were washed seven times with a 35-fold excess of IP buffer containing 0.1% DMNG and eluted with 2x SDS-PAGE sample buffer. The samples were analyzed by SDS-PAGE and immunoblotting with anti-Flag (THE DYKDDDDK Tag antibody, Genscript) and anti-V5 antibodies (THE V5 Tag Antibody, GenScript) with the inputs loaded at 2%.

## Replication

All experiments in this submission were repeated at least three times with biological replicates from independent yeast transformations.

## Acknowledgements

We thank Pedro Carvalho, Jiwon Hwang, Alex Stein, and Xudong Wu for critical reading of the manuscript. BGP was supported by the NIH/NIGMS Michigan Predoctoral Training in Genetics (T32GM007544). RDB was the Fraternal Order of Eagles Fellow of the Damon Runyon Cancer Research Foundation (DRG-2184–14). RDB is supported by the Damon Runyon Cancer Research Foundation (DFS-26–18), the University of Michigan Medical School Biological Sciences Scholars Program, and a NIH/NIGMS Award (R35GM128592). TAR is a Howard Hughes Medical Institute Investigator and is supported by a NIH/NIGMS Award (R01GM052586). The content is solely the responsibility of the authors and does not necessarily represent the official views of the NIH.

## Additional information

### Funding

| Funder | Grant reference number | Author |
|---|---|---|
| National Institute of General Medical Sciences | R35GM128592 | Ryan D Baldridge |
| National Institute of General Medical Sciences | R01GM052586 | Tom A Rapoport |
| Damon Runyon Cancer Research Foundation | DRG-2184-14 | Ryan D Baldridge |
| Damon Runyon Cancer Research Foundation | DFS-26-18 | Ryan D Baldridge |
| University of Michigan Medical School | Biological Sciences Scholars Program | Ryan D Baldridge |
| Howard Hughes Medical Institute | | Tom A Rapoport |
| National Institute of General Medical Sciences | Michigan Predoctoral Training in Genetics (T32GM007544) | Brian G Peterson |

The funders had no role in study design, data collection and interpretation, or the decision to submit the work for publication.

### Author contributions
Brian G Peterson, Morgan L Glaser, Investigation, Writing—review and editing; Tom A Rapoport, Conceptualization, Formal analysis, Supervision, Funding acquisition, Validation, Methodology, Writing—original draft, Project administration, Writing—review and editing; Ryan D Baldridge, Conceptualization, Formal analysis, Supervision, Funding acquisition, Validation, Investigation, Methodology, Writing—original draft, Project administration, Writing—review and editing

### Author ORCIDs
Brian G Peterson  http://orcid.org/0000-0001-6871-2336
Tom A Rapoport  http://orcid.org/0000-0001-9911-4216
Ryan D Baldridge  https://orcid.org/0000-0001-7158-7812

### Decision letter and Author response
Decision letter https://doi.org/10.7554/eLife.50903.sa1
Author response https://doi.org/10.7554/eLife.50903.sa2

## Additional files

### Supplementary files
- Supplementary file 1. Yeast strains used in this study.
- Supplementary file 2. Plasmids used in this study.
- Supplementary file 3. 1-way ANOVA and Tukey Honestly Significant Difference Test.
- Transparent reporting form

### Data availability
Data generated or analysed during this study are included in the manuscript.

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
