## [Decision Letter]

**Acceptance summary:**

This study reports the identification of a deubiquitinating enzyme that is involved in the process of ER-associated degradation (ERAD). Earlier work had shown that a central component of ERAD, the Hrd1 ubiquitin ligase, not only ubiquitinates substrates to be degraded, but also itself. This autoubiquitination was found to be important for activating Hrd1 for transport of ERAD substrates. What happens to Hrd1 that has become autoubiquitinated was not known. The authors now provide convincing evidence that in yeast, the deubiquitinase Ubp1 contributes to Hrd1 deubiquitination, thereby preventing its promiscuous degradation. The authors go on to provide additional details of how Ubp1 activity is regulated and the types of substrates that seem to rely on its activity for optimal degradation. The findings support a model in which the ubiquitination state of Hrd1 is carefully balanced by Hrd3 regulation of Hrd1 autoubiquitination and by Ubp1-mediated deubiquitination, which is in turn tempered by Usa1. The identification of a new factor in ERAD is an important advance that should be of interest to a range of readers of *eLife*.

**Decision letter after peer review:**

Thank you for submitting your article "Cycles of autoubiquitination and deubiquitination regulate the ERAD ubiquitin ligase Hrd1" for consideration by *eLife*. Your article has been reviewed by three peer reviewers, including Ramanujan Hegde as the Reviewing Editor, and the evaluation has been overseen by David Ron as the Senior Editor. The reviewers have opted to remain anonymous.

The reviewers have discussed the reviews with one another and the Reviewing Editor has drafted this decision to help you prepare a revised submission.

Summary:

The authors previously discovered that Hrd1 autoubiquitination is required for substrate retrotranslocation (Baldridge and Rapoport, 2016), suggesting a model in which Hrd1 functions as a ubiquitin-gated channel. The fate of poly-ubiquitinated Hrd1 was unclear, but was presumed to be de-ubiquitinated so it can be re-used for additional cycles of ERAD. The mechanism by which ubiquitinated Hrd1 is de-ubiquitinated to evade degradation was unknown. Here, Peterson and co-authors find that the membrane-tethered DUB Ubp1 deubiquitinates Hrd1, resulting in stabilization of Hrd1, but not Hrd1 substrates. Furthermore, Ubp1 is inhibited by the UBL domain of Usa1. The experiments are rigorously designed and interpreted carefully for the most part. The findings support an intriguing model in which the ubiquitination state of Hrd1 is carefully balanced by Hrd3 regulation of Hrd1 autoubiquitination and by Ubp1-mediated deubiquitination, which is in turn tempered by Usa1.

The major strength of the study is the identification of a factor that regulates Hrd1 abundance and/or activity. The quality of the data that are presented are excellent and convincing. The major drawback however is that nearly all of the experiments are performed in an over-expression paradigm under conditions with one or more Hrd1 accessory factors are deleted. Indeed, pretty much all effects of Ubp1 are really only seen when Hrd3, an integral component of the Hrd1 complex, is deleted. It was difficult to glean from the study what the actual consequence of Ubp1 deletion, mutation, or overexpression is for ERAD under otherwise unperturbed conditions, raising some doubt about whether it really plays a central role in regulating Hrd1 activity (as opposed to solely its abundance) as proposed by the authors.

Essential revisions:

The first point below was judged by the reviewers to be essential to the study and where the priority for revisions should be focused. The second and third points are also important, but less central. It is suggested that the authors try to address these experimentally, but if that is not possible, they should adjust their interpretations accordingly.

1) Nearly all of the effects that are observed on substrate degradation are seen in the context of *hrd3* with *ubp1* overexpression. While it is certainly clear that Ubp1 protects Hrd1 from degradation in this context, the evidence for a crucial role in the ERAD activity cycle is far less compelling. One of the very few experiments where Ubp1 is deleted or over-expressed in otherwise normal cells is Figure 1G. Based on what is shown, it is hard to see any effects on CPY* degradation. To the extent any effect is seen (hard to know without quantification of multiple experiments), it would appear that CPY* is degraded more effectively in *ubp1∆* cells given that there is less CPY* to start with (e.g., lane 1 vs. 5). In short, the authors only see a key role for Ubp1 under the artificial condition where Hrd3 is removed. The study requires a more thorough analysis of the Hrd1 complex and several key ERAD substrates in cells lacking or overexpressing Ubp1 in otherwise unperturbed cells. This would allow a reader to clearly judge what effect Ubp1 is having on levels of Hrd complex components, Hrd1 ubiquitination, and most importantly, rates of degradation of key ERAD substrates. If it turns out that there are essentially no obvious effects on ERAD, the central claim that Ubp1 is a key regulator of the putative ubiquitination cycle that control Hrd1 activity would seem to be overstated. In this case, a more cautious interpretation should be presented.

2) Usa1, via its UBL domain, inhibits Ubp1. It is unclear if this effect is direct or indirect. Is there an interaction between Usa1 and Ubp1? Measurement of Ubp1 activity in vitro in the presence and absence of the Usa1 UBL could be useful to support the proposed model.

3) The co-immunoprecipitation experiment shown in Figure 1H is marginal. The significance of Ubp1-Hrd1 binding could be strengthened by measuring the interaction between Hrd1 and variants of Ubp1 examined in Figure 2. Does binding of these Ubp1 variants to Hrd1 correlate with their ability to stabilize the ubiquitin ligase in the absence of Hrd3?

---

## [Author Response]

Essential revisions:The first point below was judged by the reviewers to be essential to the study and where the priority for revisions should be focused. The second and third points are also important, but less central. It is suggested that the authors try to address these experimentally, but if that is not possible, they should adjust their interpretations accordingly.1) Nearly all of the effects that are observed on substrate degradation are seen in the context of hrd3∆ with ubp1 overexpression. While it is certainly clear that Ubp1 protects Hrd1 from degradation in this context, the evidence for a crucial role in the ERAD activity cycle is far less compelling. One of the very few experiments where Ubp1 is deleted or over-expressed in otherwise normal cells is Figure 1G. Based on what is shown, it is hard to see any effects on CPY* degradation. To the extent any effect is seen (hard to know without quantification of multiple experiments), it would appear that CPY* is degraded more effectively in ubp1∆ cells given that there is less CPY* to start with (e.g., lane 1 vs. 5). In short, the authors only see a key role for Ubp1 under the artificial condition where Hrd3 is removed. The study requires a more thorough analysis of the Hrd1 complex and several key ERAD substrates in cells lacking or overexpressing Ubp1 in otherwise unperturbed cells. This would allow a reader to clearly judge what effect Ubp1 is having on levels of Hrd complex components, Hrd1 ubiquitination, and most importantly, rates of degradation of key ERAD substrates. If it turns out that there are essentially no obvious effects on ERAD, the central claim that Ubp1 is a key regulator of the putative ubiquitination cycle that control Hrd1 activity would seem to be overstated. In this case, a more cautious interpretation should be presented.

We appreciate the insight into an important deficiency in our original manuscript. We have introduced several new figure panels to address these comments. These experiments were all performed in a *hrd1Δubp1Δ* strain so we can clearly follow both Hrd1 and ERAD substrates either without Ubp1, with Ubp1, or with overexpressed Ubp1.

We have updated Figure 1G to a new figure that is the quantification of 5 biological replicates in this strain (and repositioned it to Figure 1H). In the absence of Ubp1 (Figure 1H, solid square with dashed black line), CPY* degradation is slower than with Ubp1 expressed, or overexpressed (solid black line and red line). Overexpression of Ubp1 (relative to endogenous levels of Ubp1) did not increase the degradation rates of the CPY* substrate. These data support the idea that Ubp1 is promoting optimal ERAD-L degradation. Hrd1 autoubiquitination is the primary driver of ERAD-L degradation, but deubiquitination is required for optimal efficiency (Figure 1H).

With an overexpressed, GFP-tagged version of CPY*, expression of Ubp1 improves degradation (Figure 1—figure supplement 3A, panel 3 vs. panel 2). Compared to the *ubp1Δ*, overexpression of Ubp1 also improves degradation of GFP-CPY* (panels 4 vs. 2).

With ERAD-M substrates, Ubp1 is not required for degradation beyond Hrd1 stabilization (Figure 1—figure supplement 3B, panel 2 vs. 3). In fact, expression or overexpression of Ubp1 seems to reduce the degradation of both Hmg2-GFP (Figure 1—figure supplement 3B panel 2 vs. 4) and Erg3 (Figure 1—figure supplement 3C). This is likely because ERAD-M substrates are accessible to Ubp1 on the cytoplasmic face of the ER membrane and are transferred to Dfm1 for extraction from the membrane.

Because we have now included data that support a role for Ubp1 beyond simply stabilizing Hrd1, we feel that the original interpretations are well supported. We have shown that Ubp1 helps to regulate Hrd1’s activity (primarily for ERAD-L). However, we agree there is a relatively small effect on ERAD function and have adjusted our interpretations to be more cautious and have indicated that other DUBs might be performing a similar function in the absence of Ubp1 (Discussion, fourth paragraph and model in Figure 4E)

*2) Usa1, via its UBL domain, inhibits Ubp1. It is unclear if this effect is direct or indirect. Is there an interaction between Usa1 and Ubp1? Measurement of Ubp1 activity* in vitro *in the presence and absence of the Usa1 UBL could be useful to support the proposed model.*

We agree that these experiments would be helpful to support our model. We used recombinantly expressed and purified Hrd1, Usa1, and Ubp1 and were unable to detect direct interactions (perhaps unsurprisingly). Regrettably, we do not have additional data to address this point.

3) The co-immunoprecipitation experiment shown in Figure 1H is marginal. The significance of Ubp1-Hrd1 binding could be strengthened by measuring the interaction between Hrd1 and variants of Ubp1 examined in Figure 2. Does binding of these Ubp1 variants to Hrd1 correlate with their ability to stabilize the ubiquitin ligase in the absence of Hrd3?

We have provided a new figure (Figure 2E) that demonstrates the ability of the Ubp1 variants to interact with Hrd1 roughly correlates with the effect on Hrd1 levels. We tested new Ubp1 chimeras for co-immunoprecipitation with Hrd1. Both wild-type Ubp1 and Ubp1, in which the TM segment was replaced by that of Cue4 (TM(Cue4)Ubp1 in Figure 2E) interacted weakly with Hrd1. In contrast, Ubp1 missing its transmembrane segment (Ubp1 ΔTM) did not interact with Hrd1. As expected, a soluble version of Ubp1 carrying the H-domain of Usa1 showed strong interaction with Hrd1 (Figure 2E). These data support a general correlation between Ubp1’s Hrd1 interaction and its effect on Hrd1 stabilization.